

**Use of an external organic carbon source in the removal of nitrates in Bio-sand**
**filters (BSF)**
Crispen Mutsvangwa[1] and Evans Matope[1]
[1]Department of Civil Engineering & Surveying, Cape Peninsula University of Technology, P.O. Box 1906,
Bellville 7536, Cape Town, South Africa
*Correspondence to*: Crispen Mutsvangwa (mutsvangwac@cput.ac.za)
**ABSTRACT**
Bio-sand filters (BSF) are point of use (POU) potable water filtration systems
commonly used in low-income communities at household level. The principle of
operation is similar to that of a slow sand filter and the major difference is that they are
operated intermittently at the point of use. It is one of the emerging low cost
technologies which makes use of readily and locally available construction materials
but is poor in the removal of nitrates. In order to enhance the removal of nitrates
through de-nitrification, a modified bio-sand filter with ethanol as an external carbon
source at C/N ratios of 1.1 and 1.8 was investigated. In the absence of an external
carbon source, the nitrate removal efficiency was 32% whilst removal efficiencies at
C/N ratios of 1.1 and 1.8 were 44% and 53% respectively. The inflow rate reduced
significantly from an initial flow rate of 0.04m$^3$/hr to 0.01m$^3$/hr. The reduction in the
inflow rate was mainly due to the growth of the biological layer on the filter media. The
study showed that the use of an external carbon source like ethanol in biosand filtration
enhances the removal of nitrates in potable water.
**INTRODUCTION**
Bios-sand filters are intermittent slow sand filters designed for household use and
hence called point of use (POU) water filtration systems, with principal filtration
mechanisms being physical, chemical and biological. The biological mechanisms take
place at the top layer where a biological mat develops in the 50 to 100mm of the media.
The biological layer acts both as a fine filter to remove small colloidal particles,
dissolved impurities and at the same time immobilizes pathogens.
Although bio-sand filters are now widely applied in the treatment of water at household
level, few studies have been conducted on the removal of chemical contaminants.
Current research in BSF has mainly focussed on the removal of pathogenic organisms
like *Escherichia coli (E. coli)* and suspended solids. One chemical of major concern
is nitrate-nitrogen contamination ($NO_3$-N) in surface and ground water as it poses
serious health problems (Almasiri and Kaluarachchi, 2007). Methaemoglobinemia in
infancy is related to nitrate ingestion resulting in low oxygen in-intake and
consequently causing death. Furthermore, presence of nitrates in drinking water
results in the formation of nitrosomines in the stomach, which are carcinogenic. Nitrate
poisoning has been reported in livestock when concentrations exceeded 100mg/l
(Tredoux et al., 2000) and other problems related to nitrate in drinking water are well
documented in literature (Moraes 1995; Fan and Steinberg, 1996; Lin et al., 2002;
Forman 2004).
Main sources of $NO_3$-N in surface waters and groundwater aquifers include use of
agricultural fertilizers, animal waste disposal, wastewater effluents from conventional
and on-site sanitation facilities. Water supply from high nitrate concentration



environments needs some form of treatment or dilution with low-nitrate content water.
The current design of conventional biosand filters has been proved to be poor in the
removal of nitrates (Heather et al., 2010; Mahlangu et al., 2011; Kennedy et al., 2012).
Physical and chemical methods such as ion-exchange, reverse osmosis, electro-
dialysis, distillation, nanofiltration and activated carbon have been applied in the
removal of nitrates from drinking water supplies. These methods are relatively
expensive and show poor selectivity for nitrate removal with generation of brine, which
is difficult to dispose (Moheseni et al., 2013). Hence there is need to explore
alternative technologies like biological de-nitrification which has been proved to be
efficient in complete nitrate elimination and has the advantage of producing a harmless
by-product ($N_2$). The pathway for nitrate removal by heterotrophic bacteria is:-nitrate
→nitrite→ nitric oxide → nitrous oxide →gaseous diatomic nitrogen:-

$$NO_3^- \rightarrow NO_2^- \rightarrow NO \rightarrow N_2O \rightarrow N_2 \uparrow$$

The biological de-nitrification technology is based on the conventional theory, that
carbon is the limiting factor in the efficiency of biological de-nitrification. Heterotrophs
utilises carbon from organic compounds like sugars, organic acids and amino acids as
source of electrons rather than from inorganic compounds like carbon dioxide as the
case in autotrophic de-nitrification. Although autotrophic nitrate removal has the
advantage of not requiring an organic carbon source, it is associated with slow growth
rate of autotrophic bacteria and low nitrate removal rate.
Few studies have been conducted on the ability of the bio-sand filters in the removal
of nitrates. In a study conducted in rural Cambodia by Heather et al. (2010), it was
revealed that there was simultaneous nitrification and de-nitrification occurring in the
bio-sand filters. However, about 85% of the biofilters under study did not meet the
WHO guideline for $NO_3$-N in the treated effluent. The study showed that de-nitrification
was predominant when the inflow into the filter was from surface water, which could
be due to the high organic carbon content. Kennedy et al. (2012) studied the effects
of hydraulic loading on removal of nitrates in biosand filters and the overall nitrate
removal efficiency was low (16%). Mahlangu et al. (2011) established that the
conventional BSF and the modified BSF of zeolites (clinoptilote) have relatively low
removal rates of nitrates (37%). On the same study, other types of biofilters which
include ceramic candle and bucket filters had poor removal of nitrates ranging from
18% to 37 %. On certain occasions, the effluent concentration of $NO_3$-N was even
higher than the unfiltered water and possibly due to desorption of previously adsorbed
nitrates.
Most sources of drinking water lack sufficient quantities of organic carbon for cell
growth and energy source. The organic carbon acts as both a source of cellular
material for biological respiration and electron donor for dissimilatory nitrate reduction.
Waters with low carbon content require an external carbon source for de-nitrification
to take place under anoxic conditions and nitrate is converted to gaseous diatomic
nitrogen.
A variety of external carbon sources like sucrose, ethanol, methanol and acetic acid
have been applied in conventional slow sand filters to aid heterotrophic denitrification
at C/N ratios ranging from 1 to 2.5. The studies have shown considerable
improvement levels in the de-nitrification process due to the recorded high nitrate





removal efficiencies of about 90% (Green et al., 1994).  Gomes et al. (2000) assayed
the influence of sucrose, ethanol, methanol and ethyl alcohol in nitrate reductase in
contaminated groundwater and showed very high removals with effluent
concentrations ranging from 0 to 5mg/l.  Aslan and Cakici (2007), reported removal
rate of 94% for nitrate in slow sand filters when acetic acid was used as a carbon
source.  Methanol is toxic due to some of the residual concentrations of carbonaceous
compounds found in the effluent and produces an excessive growth of biomass.
Sucrose and glucose have a tendency to form a biomass which increases turbidity in
the final effluent.  Acetic acid and ethanol are considered to be the most suitable
carbon sources in removal of nitrate and no limits have been set in potable water.
They are also cheaper, a concept inherent with the use of bio-sand filtration
technology.
However, heterotrophic de-nitrification has not been investigated in bio-sand filters
except in the conventional slow sand filters.  The aim of this study was to investigate
the removal of $NO_3$-N in biosand filters with ethanol as a carbon source and to
establish the optimum Carbon to Nitrate (C/N) ratio for microbial activity which
achieves maximum removal with minimum excess carbon in the effluent.
**MATERIALS AND METHODS**
Two bio-sand filters were investigated at household level: - one with an external
carbon source (BSFC) to enhance the de-nitrification process at C/N ratios of 1.1 and
1.8, and the other one without a carbon source (BSF).  The two bio-sand filters were
dosed with known concentrations of ammonium nitrate which was the source of nitrate.

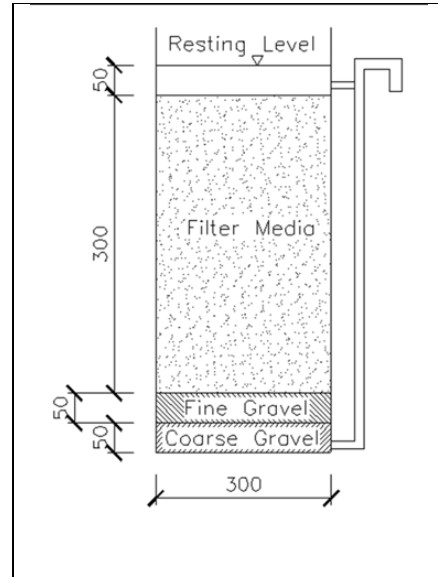

Figure 1        Schematic representation of the bio-sand filter (dimensions in mm).
**Filter construction**
The two bio-sand filters were constructed according to the Centre for Affordable Water
and Sanitation Technology guidelines (CAWST 2008).  Plastic buckets 25ml in volume



were used and were packed with multi-media filter material.  The multi-media filter bed
consisted of fine sand of 0.3mm diameter and 250mm deep; sand of 0.95mm diameter
and 750mm deep; gravel of 7mm diameter and 50mm deep.  The South African
National Standard (SANS 3001) were used to determine the particle size and grading
in order to achieve the required particle size distribution of the filter media.  Dewatering
of the filter between charges is avoided by a vertical discharge tube that rises from 2-
7 cm above the height of the filter media.  The elevated outlet allows the media to
remain saturated after a charge has been filtered and when water is no longer flowing
from the outlet (Fig. 1).  The design parameters of the filter are summarised in Table
142    1.
Table 1: Summary of the design values used for the two filters (BSF & BSFC)

| Design parameter | Unit | Recommended Value | Applied Value |
|---|---|---|---|
| Media depth | m | 0.3-0.5 | 0.3 |
| Supernatant depth | mm | 50 | 50 |
| Surface area | $m^2$ | 0.06 | 0.071 |
| Effective size | mm | 0.15-0.40 | 0.35 |
| Coefficient of uniformity | --- | 1.5 to 3 | 2.64 |
| Filtration velocity (in clean filter bed) | m/hr | 0.10 to 0.6 | 0.17-0.63 |
| Inflow rate | $m^3$/hr | 0.03 to 0.04 | 0.04 |

The filtration cycle of a biofilter is made-up of resting time (6-24hrs) and a maximum
filtration time of about 2hrs (Fewster et al., 2004).  The biological treatment occurs
during the resting time and after this period the filter bed is drained.  In this study the
raw surface water was fed into the filter once a day and the resting time and filtration
time were 24hrs and 2hr respectively.  The filtered water was collected in a 5 litre
vessel for laboratory analysis.  The average inflow rate was measured from noting the
start time of filtration and the time periods at which the level of the water in the
receiving vessel changed by 1 litre.
The superficial velocity ($v_s$) is related to the surface area of the filter and is normally
used in filtration computations and is also equivalent to the hydraulic surface loading
divided by the surface area of the filter.  For BSF, the inflow rate is not constant since
the water is only poured once for a filter cycle and hence the infiltration velocity
decreases with time from the start to end of cycle.
**Nitrate and Carbon source dosage**
A stock solution of ammonium nitrate ($NH_4NO_3$) of concentration of 190g/l was dosed
to both filters (BSF and BSFC), and to achieve a dose of 25mg/l in the 25l filter volume,
3.33ml of the stock solution was required.  The ethanol was applied only to BSFC at
C/N ratios of 1.1 and 1.8.  With a molar mass of 46g/mol of ethanol ($C_2H_5OH$) the
carbon equivalent in the ethanol was 24g/mol (52.2%).  Therefore, at a nitrate dose of
25mg/l and C/N ratio of 1.1 the dosage of carbon as ethanol in a 25l biosand filter was
7.45ml of carbon as ethanol.  Similarly, at C/N ratio of 1.8, the required dose of carbon
as ethanol was12.1ml.  The surface loading of $NO_3$-N was calculated by multiply the
concentration of nitrate with the superficial velocity (g/$m^2$.d).



**Filter maturation**
The de-nitrification in biosand filters is biological and take place under a fixed film
growth process whereby the bacteria develop on the surface of the sand media. For
the smooth operation of the biosand filter, the water level was maintained at 50mm
above the fine sand. The maturation period for the full development of the biological
layer and acclimatising of the microorganisms to ethanol and $NO_3$-N environment was
3 weeks. The operating temperatures of the filters varied between 19°C and 20°C and
were not controlled in order to simulate the actual operating conditions of a biosand
filter at household level.

**Sample Collection and Analysis**
Sampling bottles were washed with distilled water before and after sampling. The
samples were collected at the inlet and outlet of the two biosand filters in 500ml
Erlenmeyer flasks and stored in a refrigerator at $4^0$C and analysed within 1 hour. The
frequency of sample collection was once a week after the 12 hr resting time.

The pH and dissolved oxygen (DO) were measured using a pH meter Model HACH
HQ30D (FLEXI Model). The instrument was calibrated and measurements conducted
in accordance with the Standard Method. The nitrate was measured by Spectroquant
Nitrate Photometrical Test Method using Merck Spectrophotometer PHARO100 and
the results were reported as $NO_3$-N in mg/l. The carbon source which was ethanol
was measured as Chemical Oxygen Demand (COD) by the MERCK Spectroquant TR
320 Digester (Spectroquant COD Cell Test method). The samples were digested in
tubes containing a mixture of chromic and sulphuric acid with silver sulphate as a
catalyst. After digestion samples were cooled and read on the Spectroquant
PHARO100 Spectrophotometer. The COD test was carried out mainly to determine
the amount of ethanol as a carbon source in the source water before and after the
filtration process.
**RESULTS AND DISCUSSIONS**
**Flow rates**
Initial flow rates in the control filter BSF started from 0.04m³/hr and declined to
0.03m³/hr by end of experiment, whilst in BSFC which received carbon source the flow
rate reduced from 0.04m³/hr to 0.01m³/hr (Fig. 2). The reduction in flow rates was
comparable to studies conducted on Bio-sand filters by Kubare & Haarhoff et al.,
(2010) and Kennedy et al., (2012). The declining in the filtration rate was due to filter
clogging and was substantial when the biological layer was fully mature. The reduction
in the flow rate was more pronounced in the filter dosed with an external carbon source
(BSFC) compared to one without carbon (BSF). Therefore, there was more growth of
the biomass in the biofilter with an external carbon source due to the favourable
environment conducive for growth of heterotrophic bacteria. Conventional surface
cleaning will not remove the biomass at the bottom layers. Consequently, a household
would require more filters to meet the daily water demand as well as increasing the
resting period in BSFC to reduce excessive growth of biomass. Overall, the filtration
velocity ranged from 0.17m/hr to 0.63m/hr and typical filtration rates for BSF range
from 0.16 to 1.1m/hr (Elliot et al. 2008; Kubare & Hannoff 2010).





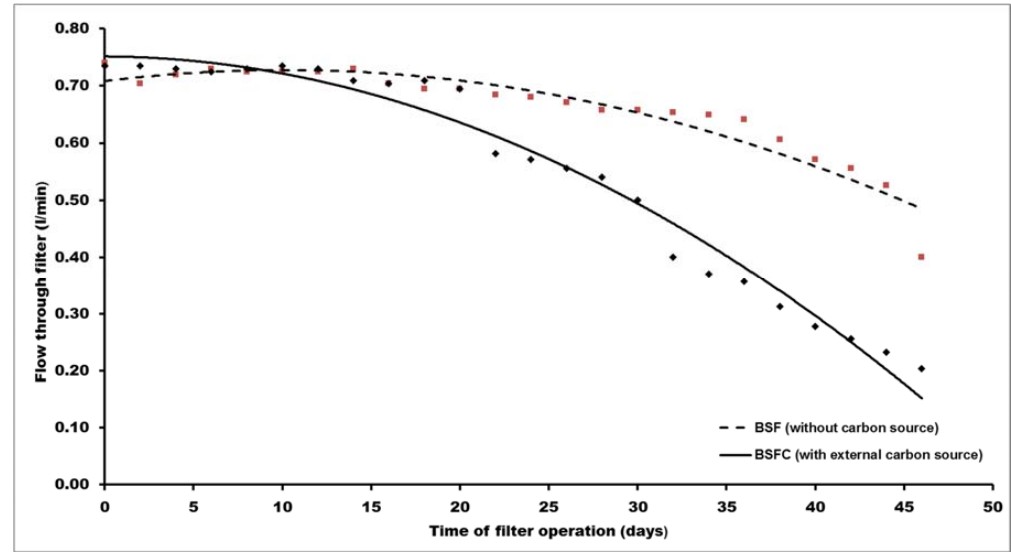

Figure 2:      Variation of flow rates in the filters with and without carborn source

**Changes in pH and DO**
The pH and DO are important physiochemical parameters in the removal of nitrates in BSF.  There was no significant change in the pH of the influent and effluent water for both filters (BSF and BSFC).  Overall, there was a slight decrease in pH fron 8.6 to 6.8 and such a pH range would favour the de-nitrification process since maximum de-nitrification rates are acheived at pH range of 7 to 8.5.  Whereas for pH values smaller than 6 and larger than 8.5 would result in a sharp decrease in the de-nitrification activities.  However pH may increase during de-nitrification because the reduction of nitrate to gaseous nitrogen with organic substrate as an electron donor results in the production of carbon dioxide and oxygen hydroxide ($OH^-$), which may react to form a bicarbonate ($HCO_3^-$) and carbonate ($CO_3^{2-}$) (Drtil et al. 1995; Wang et al., 1995).  With regards to water quality guidelines, the pH values were within the acceptable South African (2015) guideline limits of 5.0 to 9.7.

The overall reduction of DO in the filter with an external carbon source was 65% with average inflow and outflow concentrations of 8.23mg/l and 2.94mg/l respectively.  However, the reduction in dissolved oxygen was less in the filter without an external carborn source (50%).  The reduction in the DO is due to the oxgen demand by aerobic and nitrifying bacteria at the top of the filter bed.

**Nitrate Removal Rates**
The nitrate removal mechanisms during heterotrophic de-nitrification are bacterial respiration and bacterial synthesis (Mohseni-Bandp et al., 2013).  The de-nitrification will take place at the bottom of the filter bed where there is less oxygen (anoxic conditions).  William and Beresford (1986) concluded that nitrification and de-nitrification happen simultaneously in zones where there are short distances between the aerobic and anaerobic zones.  The same scenario is depicted in biosand filters due to the short filtration length of apprximately 0.3-0.5m (Elliot, CWAST).



Heterotrophic bacteria need organic carbon as the electron donor and as the source
of carbon, whilst getting their oxygen by removing bound oxygen from nitrate ($NO_3^-$)
which is in the water being treated and the nitrate acts as the electron acceptor. As a
result of this process, the removal rate of nitrates in the filter without external carbon
source (BSF) was 30%±0.04 (Table 2) and Mahlangu et al. (2011) reported a rate of
37% in similar filters. In the filter with an external carbon source (BSFC) the nitrate
removal rate was 44%±0.03 at C/N ration of 1.1 and 53%±0.02 at C/N ratio of 1.8.
Overall the nitrate removal rate was higher with the use of an external carbon source
at higher C/N ratio of 1.8 (Table 3). However, the effluent nitrate concentration was
between 16 to 19mg/l but still above the recommended guideline values in potable
water.
The failure to achieve effluent nitrate guideline values even though pH was optimum
could be due to high DO. Optimum de-nitrification occurs under anoxic conditions
when oxygen levels are depleted (low redox) and nitrate becomes the primary oxygen
source for heterotrophic bacteria. In general it has been observed that a dissolved
oxygen concentration of more than 0.2mg/l reduces the rate of de-nitrification
significantly (Jorgensen and Sorensen, 1984). High levels of dissolved oxygen were
recorded ranging between 2.9mg/l to 8.2mg/l and hence were higher than the optimum
values for de-nitrification.
Reducing the DO concentration in biosand filter will enhance the nitrate removal
efficiency but will compromise the aerobic microbial activity at the top layer. A feasible
alternative will be to increase the filter depth so as to create an anoxic zone at the
bottom or to increase the resting period of the filter. Bio-sand filters are designed with
a filtration time of 2 hrs and resting period of 12 to 24 hours (CAWST, 2007; Elliott et
al., 2008). The resting time provides the contact time for microbial removal and de-
nitrification processes and thus a long resting time will be desirable from this
perspective. However, too long a resting period may reduce the viability of the
biological layer because the survival of the microorganisms relies on the periodic
inflow of source water for nutrients (Baumgartner et al., 2007). Additionally, too long
a resting period will reduce the water production rate and thus fail to satisfy household
water requirements. Therefore careful selection of the resting period is vital in order
to balance these competing objectives. In this study a resting time of 12hrs was used
and nitrate concentrations measured during this period showed a rapid removal rate
during the first 1.5 hrs and no significat removal thereafter (Fig. 3). Therefore,
increasing the resting period more that the 12 hours will not have any sigificance in
the nitrate removal. However, results for the entire operational period indicates low
removal at the begining (40%) and thereafter the rate increased to 53%, and this
illustrates the importance of maturation period. The variation of nitrate concentrations
for the entire operational period are illustrated in Fig. 4.

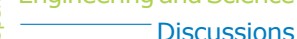

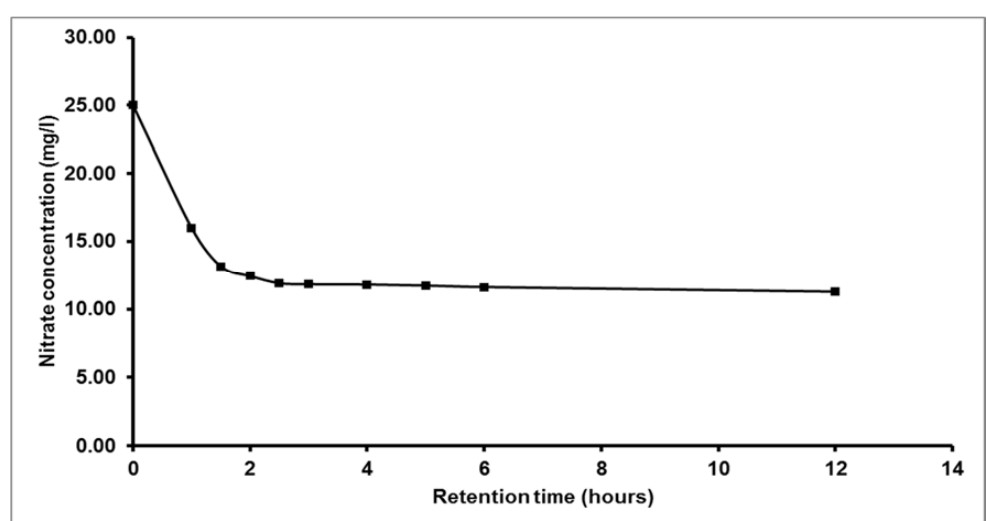

Figure 3: Reduction of nitrate relative to resting period in the filter with an external
carbon source. Values of the nitrate are the average of the C/N ratio of 1.1 and 1.8.

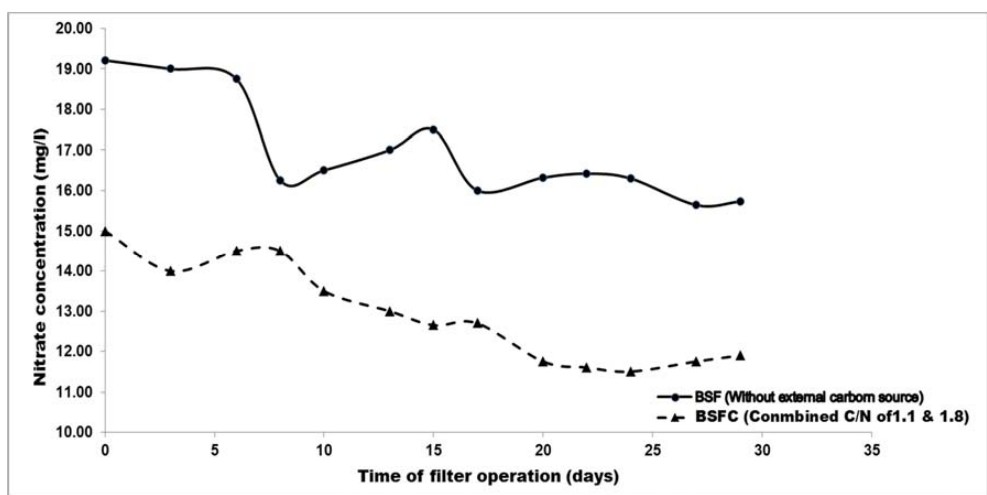

Figure 4:      Variation of nitrtate concentrations for the entire operational period
**Denitrification Rate**
The denitrification rate was computed as:

$R_{dn} = \dfrac{1}{t}(C_{in} - C_{out})$
Where:-
$R_{dn}$      = denitrification rate (M/L$^3$T)
$C_{in}$      = influent nitrate (M/L$^3$)
$C_{out}$     = effluent nitrate (M/L$^3$).



The denitrification rate for BSF and BSFC were 3.66gNO$_3$-N/m$^3$.d and 5.44gNO$_3$-
N/m$^3$.d respectively and these rates are lower than those reported by Aslan et al.,
(2007) in slow sand filters ranging between 8.1 and 29.2 gNO$_3$–N/m$^3$.day at filtration
rates between 0.015 and 0.06 m/h.
Table 2:      Nitrate removal effficiency at C/N=1.1 ant at influent nitrate concentration
of 25 mg/l

| Sampling interval (Days) | BSF (No external Carbon Source) | | BSFC C/N=1.1 (With external carbon source) | |
|---|---|---|---|---|
| | Effluent Nitrate (mg/l) | Removal Efficiency | Effluent Nitrate (mg/l) | Removal Efficiency |
| 0 | 19.21 | 23% | 15.00 | 40% |
| 2 | 19.00 | 24% | 14.00 | 44% |
| 5 | 18.75 | 25% | 14.50 | 42% |
| 7 | 16.25 | 35% | 14.50 | 42% |
| 9 | 16.50 | 34% | 13.50 | 46% |
| 12 | 17.00 | 32% | 13.00 | 48% |
| 14 | 17.50 | 30% | 12.65 | 49% |

Table 3:      Nitrate removal effficiency at C/N=1.8 ant at iinfluent nitrate
concentration of 25 mg/l

| Sampling interval (Days) | BSF (No external Carbon Source) | | BSFC (With external carbon source) C/N=1.8 | |
|---|---|---|---|---|
| | Effluent Nitrate (mg/l) | Removal Efficiency | Effluent Nitrate (mg/l) | Removal Efficiency |
| 17 | 16.00 | 36% | 12.70 | 49% |
| 20 | 16.32 | 35% | 11.75 | 53% |
| 22 | 16.42 | 34% | 11.60 | 54% |
| 24 | 16.30 | 35% | 11.50 | 54% |
| 27 | 15.64 | 37% | 11.75 | 53% |
| 29 | 15.73 | 37% | 11.90 | 52% |

**Residual COD in effluent**
The residual ethanol measured as COD in filters with an external carbon source varied
between 25mg to 35mg/l.  Overall, the removal efficiency of COD at C/N ratio of 1.1
and 1.8 was 89% and 90% respectively.  There was rapid COD removal in the first 2
hours and became constant as the resting period increases and hence there is no
significant benefit with longer resting periods.  The same trend is depicted with nitrate
removal which concluded that the de-nitrification process takes place in the first 2
hours when the COD is utilised in the process.  However, the COD concentrations in
the effluent were higher than the guideline values, and such high level of COD
concentrations may be toxic to human health and increases disinfection by-product
formation potential.  This present a major health challenge in the use of an external

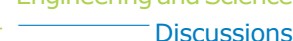



carbon source in the removal of nitrates in potable water and there is a need to explore
post-treatment methods to remove the residual carbon in biosand filters.

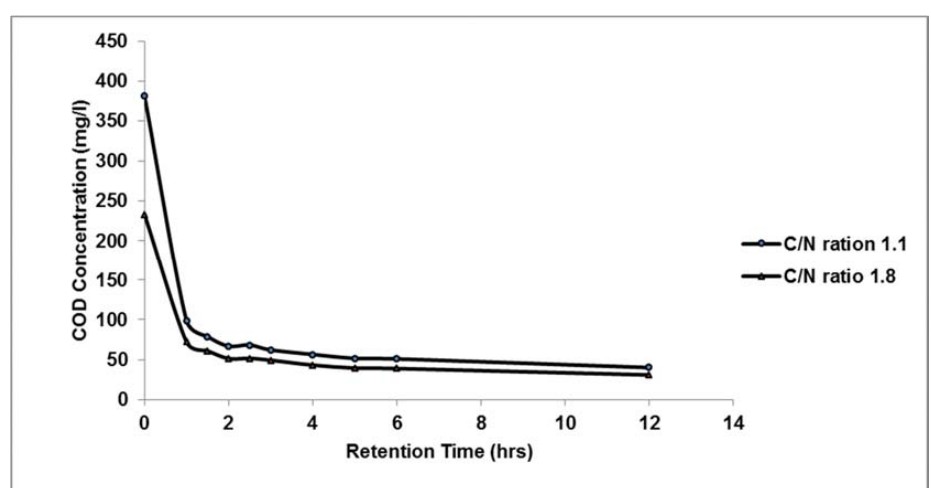

Figure 5: Reduction of COD relative to resting period in the filter with an external
carbon source
Table 4:      COD removal Efficiency at C/N =1 and C/N=1.8

| Sampling interval (days) | Influent COD (mg/l) | Effluent COD (mg/l) | COD Removal Efficiency (%) | Sampling interval (days) | Influent COD (mg/l) | Effluent COD (mg/l) | COD Removal Efficiency (%) |
|---|---|---|---|---|---|---|---|
| | C/N = 1.1 | | | | C/N = 1.8 | | |
| 0 | 233.52 | 26.85 | 88.50 | | | | |
| 2 | 233.52 | 25.30 | 89.17 | | | | |
| 5 | 233.52 | 25.61 | 89.03 | | | | |
| 7 | 233.52 | 23.98 | 89.73 | | | | |
| 9 | 233.52 | 24.77 | 89.39 | | | | |
| 12 | 233.52 | 25.10 | 89.25 | | | | |
| 14 | 233.52 | 26.36 | 88.71 | | | | |
| 17 | 382.12 | 35.54 | 90.70 | 17 | 382.12 | 35.54 | 90.70 |
| 20 | 382.12 | 36.10 | 90.55 | 20 | 382.12 | 36.10 | 90.55 |
| 22 | 382.12 | 35.86 | 90.62 | 22 | 382.12 | 35.86 | 90.62 |
| 24 | 382.12 | 35.42 | 90.73 | 24 | 382.12 | 35.42 | 90.73 |
| 27 | 382.12 | 35.40 | 90.74 | 27 | 382.12 | 35.40 | 90.74 |
| 29 | 382.12 | 35.18 | 90.79 | 29 | 382.12 | 35.18 | 90.79 |

**CONCLUSIONS**
Bio-sand filtration enhanced by an ethanol as an external carbon source has potential
in the removal of nitrates in potable water at household level.  The average nitrate
removal efficiency in biosand filter with ethanol as an external sourc at C/N

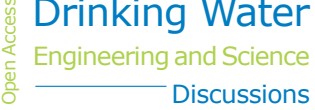

ratios of 1.1 and 1.8 was 44% and 53% respectively.  Although the nitrate
concentration levels in effluent exceeded the recommemded guidelines, the
technology is capable of limiting nitrate in drinking water.  Increasing the resting period
more that than 12 hours will not have any sigificance in the nitrate and COD removal
rates since these two processe take place in the first 2 hours.  Dissolved Oxygen
concentration in the effluent reduced significantly due to aerobic oxidation and
nitrification which took place, simultaneously on the upper layer of the filter.  The
reduced DO low levels  promoted hetertrophic de-nitrificaton at the bottom of the
biosand filter.  However, the DO levels were still above for optimum values for de-
nitrification, and also the residual COD concentrations were above the water quality
guidelines.
The flow rates reduced with time throughout the whole experiment due to the growth
of the biological layer and clogging of the filter media and as a result the yield of the
biofilter was reduced.  The flow rate reduction was higher in the filter with an external
carbon source and was substantial when the biological layer was fully mature.  Overall,
the study concluded that there is high potential in the use of POU filters enhenced with
an external carbon source in the removal of nitrates through heterotrophic
denitrification.  The major challenge on the use of an external carbon source is the
high residual COD concentration, which may pose a health risk.

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
