# Peer review of "Use of an external organic carbon source in the removal of nitrates in Bio-sand filters (BSF)"

_Drinking Water Engineering and Science, 2017_

## Referee Comment (RC1) · N. A. F. M. Kamil (Referee) · 19 Aug 2017

General comments Good Abstract. Good discussion for each element. However, required extra discussion why C/N ratio 1.8 was better than C/N ratio 1.1. Good recommendation for future research (Line 275-278)

Scientific questions/issues Line 150: Define 'raw surface water'. Line 179: How do you determine the duration for maturation period (3 weeks)? Line 346: Table 4: Need explanation why C/N ratio 1.8, no data for COD removal between 0 to 14 days.

Technical issues Line 39 & 49: Grammatical error. Change 'in' to 'on'. Line 28-33;

[Figure]

37-38; 40-42; 67-73; 87-89; 91-96; 98-100; 228-232: Include references. Line 55-57: Include references: Shaharudin, N., Suradi, N., & Kamil, N. A. F. M. (2017). Measurement of Water Quality Parameters for Before and After Maintenance Service in Water Filter System. In MATEC Web of Conferences (Vol. 103, p. 06006). EDP Sciences. This study use reverse osmosis to remove nitrogen Line 78: Grammatical error. Include 'the'. Under 'the' study. Line 91: Grammatical error. Include 'are'. Line 125: Include 'Figure 1 shows the schematic diagram of….' Line 145: Table 1: Include references for 'Recommended Value' Line 146: Grammatical error. Change 'is' to 'was' Line 174: Include word 'process': biological process Line 306-311: The equation should include in Methodology, not in result section.

---

## Referee Comment (RC2) · Anonymous Referee #2 · 21 Aug 2017

**REFEREE COMMENT**

Manuscript dwes-2017-26
Title: **"Use of an external organic carbon source in the removal of nitrates in Bio-sand filters (BSF)"**
Authors: **Crispen Mutsvangwa and Evans Matope**

In order to find efficient and low cost external carbon source for denitrification bacteria, this paper evaluates the use of the ethanol at C/N ratios of 1.1 and 1.8. The experiments were carried out in biofilters with sand fulfillment.

**GENERAL COMMENTS**

1. It is well known that the rate of denitrification process depends, inter alia, on the type and concentration of organic carbon present in water. Literature review should be more comprehensive.

2. Why do the authors focus only on measuring nitrates in the effluent? The denitrification process takes place in several stages and can stop at the formation of nitrites or other toxic by-products. Nitrites can be easily measured using conventional methods or Spectroquant Photometrical Tests.

3. Excessive amount of C relative to N can facilitate the reduction of sulfate to sulfide, a highly toxic product for fish. Have the authors considered the impact study of sulphates on the denitrification process?

4. During only 46 days of experiment with external source of carbon the flow rate decreased by 75%. The increase of microorganism biomass caused the filter clogging, so it is reason that more common is using of fluidized sand biofilters.

**SPECIFIC COMMENTS**

1. Why did the authors study at an influent nitrate concentration of 25 mg/l, if the acceptable value is 11 mg $N-NO_3^-$/ l (48.7 mg $NO_3^-$) [South African National Standard, Drinking Water 2015].

2. How can the authors explain such a high oxygen content in influent at 8.2 mg/l?

3. The authors noted a slight decrease in pH from 8.6 to 6.8. But the properly conducted denitrification process causes an increase in pH.

**TECHNICAL CORRECTIONS**

1. Please mark the direction of flow in Figure 1.

2. In Table 4, after "17 Sampling interval (days)" data are repeated for C / N = 1.1 and C / N = 1.8.

---

## Author Comment (AC1) · 11 Sep 2017

**Response to Reviewer 1**

1.  Line 149:-  The raw water has been defined by adding "untreated river water":- The statement now reads as follows:

    In this study the raw surface water or untreated river water was fed into the filter once a day and the resting time and filtration time were 24hrs and 2hr respectively

2.  Three weeks were selected based on typical maturation periods.  The following statement has been added after line 179:-

    "The biological layer typically takes 20 to 30 days to develop to maturity in a new filter depending on the quality of the inlet water (CAWST, 2009; Mahlangu, 2011)".

3       A C/N ratio of 1.8 was selected based on studies conducted by Aslan et al (2007), Gomez et al (2000) and Callado (2001).  The following statement was added:-

    "These two ratios were selected based on the optimum range of carbon to nitrogen ratio which was established by Aslan et al (2007), Gomez et al (2000) and Callado (2001) for de-nitrification in slow sand filtration, which ranged from 1.08 to 2.5".

4       Data sets corrected due to typing error

5       Technical issues: Line 38 & 49 grammatical error corrected

6       The following references have been included:-
        Lines 28-33         Murphy et., 2010 and CAWST, 2009.

        Lines 37-38         Elliott et al., 2008; Mwabi et al, 2012; Van Halem et al., 2009.
        Lines 40-42         Aslan et al., 2007; Craun et al., 1981; Shuval et al., 1977).

        Lines:-87-89        Mahlangu et al., 2011

        Lines:-91-96        Mohseni-Bandpi et al., 2013

        Lines 98-100        Aslan et al., 2007; Gomez et al., 2000 and Callado, 2001

        Lines 228-232       Wang et al., 1995; Drtil et al., 1995

        Lines 55-57         Shaharudin, et al., 2017); Shuttle et al., 2006; Shoeman et al., 2002

7       Line 78; 91;        Gramatical errors corrected

8    Line 125 corrected: "Figure 1 shows the schematic diagram of the biosand filters which were constructed"

9    Line 145:    Table 1: References included:-

Table 1: Summary of the design values used for the two filters (BSF & BSFC)

| Design parameter | Unit | Recommended Value | Applied Value |
|---|---|---|---|
| Media depth | m | 0.3-0.5
CAWST (2007); Kubare et al. (2010) | 0.3 |
| Supernatant depth | mm | 50
Lukacs (2002); Duke et al. (2006); CAWST (2007) | 50 |
| Surface area | $m^2$ | 0.06
CAWST (2007) | 0.071 |
| Effective size | mm | 0.15-0.40
CAWST (2007), Manz et al. (1993) | 0.35 |
| Coefficient of uniformity | --- | 1.5 to 3
Elliot et al. (2008); Manz et al. (1993) | 2.64 |
| Filtration velocity (in clean filter bed) | m/hr | 0.10 to 0.6
Kubare et al. (2010); Elliot et al. (2008) | 0.17-0.63 |
| Inflow rate | $m^3$/hr | 0.03 to 0.04
CAWST (2007) | 0.04 |

10    Line 146: Gramatical error corrected

11    Line 174 the word "process" included

---

## Author Comment (AC2) · 11 Sep 2017

**Response to Reviewer 2**

**General Comments**

**Comment 1**
This aspect is addressed in lines 98 to 113, where different carbon sources are discussed.

**Comment 2**

The objective of our research was to evaluate the removal of nitrates in a biosand filter to meet the water quality requirements, and not at various levels or depths. However, this could be a potential area for future research.

**Comment 3**

The objective of the study was to evaluate a biosand filter for potable use with regards to nitrates and hence excluded the negative impacts on the aquatic environment like fish.

**Comment 4**
We also appreciate this line of research of using fluidized sand biofilters. However, the additional costs associated with pumping to maintain bed expansion or fluidization must also be considered as compared to the conventional biosand filter. Hence, biosand filters are suitable for low income communities, a concept inherent with their use.

**Specific Comments**

**Comment 1**
If we had started with 11mg/l, then we were not going to be able to evaluate the removal rates above the acceptable value for potable use (narrow range). Moreover, most waters of concern have nitrate concentrations above the 11mg/l.

**Comment 2**
Dissolved oxygen concentration is influenced by a number of factors including water temperature, organic matter, salinity and atmospheric pressure. The operating temperature of the filters was between 19$^{o}$C and 20$^{o}$C (Line: 179) and the measured DO values are typical at such temperatures. Furthermore, the water which was used was raw river water and the DO can range between 0 to 18mg/l in such waters depending on level of pollution.

**Comment 3**
This phenomenon is explained in lines 232 to 237. Also other researchers have noted insignificant pH changes in similar filters (Baba et al., 2015). Another reason could be the short resting period of 24hrs because significant changes of pH are noted after 5

days (Rust et al., 2000).  Also, a higher concentration of the carbon source can result in an increase in pH.

Furthermore, It should be noted that in a biosand filter, the process is not purely de-nitrification.   There is also nitrification and aerobic respiration at the top due to availability of oxygen and this phenomenon has been explained in lines Lines 30;  32); 75-78; 246-252 and also confirmed by Heather et al. (2010) and Willian et al. (1986). Nitrification is obligatorily coupled to oxygen consumption and has an effect on the decrease in alkalinity. Such a decrease in alkalinity might cause a decrease in pH (Habboub, 2007).  Acidic nitrite formation results in a drop in pH, thus if the buffer capacity of the system is weak, the pH might drop well below 6.7

**Technical corrections**

1. Direction of flow:-Noted

2. Data sets corrected due to typing error

---

## Referee Comment (RC3) · Anonymous Referee #2 · 15 Sep 2017

Thanks for explanation. I look forward to future research on the need to analyze also nitrites. The authors themselves have observed a decrease in pH, which attests to the acidic nitrite formation. Nitrite is an unstable form, but their presence in water is undesirable and the permissible concentration is determined by the relevant Regulation.

---

## Referee Comment (RC4) · Anonymous Referee #3 · 28 Sep 2017

I would like the author to address the following; 42 - 43 - Reference (On Carcinogenic effect) 71 -73 – Reference 107 – 111 - Reference 87-89 - Apart from desorption as source of effluent nitrate rise, any contribution from death of Heterotrophs and nitrification process 123 – Why C/N ratio of 1.1 to 1.8 132 - 135 – Recheck the mentioned diameter if at all match with those shown in Figure 1 199 – 200 – COD test, what was the COD value for the raw water before ethanol feeding as the carbon source (i.e. what was the nature of raw water) 228 – 229 – Any comment on pH decrease (i.e from 8.6 - 6.8)

26, 2017.

---

## Author Comment (AC3) · 3 Oct 2017

**Response to Anonymous Referee # 3**

The following references have been included:-

    Lines 40-43        Aslan et al., 2007; Craun et al., 1981; Shuval et al., 1977.
    Lines:-71-73      Mohseni-Bandpi et al., 2013
    Lines: -107-11    Stouthamer, 1992; Cherchi et al., 2009; Jensen et al, 2012
    Lines:-87-89     Mahlangu et al., 2011

Line 87-89: Comment acknowledged and statement revised as follows:

On certain occasions, the effluent concentration of $NO_3$-N was even higher than the unfiltered water and possibly due to desorption of previously adsorbed nitrates and nitrification. Also research has revealed heterotrophic nitrifying microorganisms are key players in the nitrogen cycle and hence can also increase the effluent concentration of $NO_3$-N through cell-lysis (Masahito et al., 2007).

Aslo asked by reviewer 1 and was addressed as follows in the reply-

A C/N ratio of 1.8 was selected based on studies conducted by Aslan et al (2007), Gomez et al (2000) and Callado (2001). The following statement was added:-

"These two ratios were selected based on the optimum range of carbon to nitrogen ratio which was established by Aslan et al (2007), Gomez et al (2000) and Callado (2001) for de-nitrification in slow sand filtration, which ranged from 1.08 to 2.5"

Line 132-135
The filter diameter is 300mm as shown in Fig 1 and the diameter of the filter media as described in lines 134-135.

Nature of raw water
The COD of the raw river water was very low to be effective as a carbon source and was 24mg/l and hence the spiking. The average pH of the raw water was 8.6 (Line 228) and a nitrate concentration ranging from 0.39mg/l to 1.15mg/l.

Lines 228-229 (asked also by Reviewer # 2 and was addressed as follows in the reply:-

This phenomenon is explained in lines 232 to 237. Also other researchers have noted insignificant pH changes in similar filters (Baba et al., 2015). Another reason could be the short resting period of 24hrs because significant changes of pH are noted after 5 days (Rust et al., 2000). Also, a higher concentration of the carbon source can result in an increase in pH.

Furthermore, It should be noted that in a biosand filter, the process is not purely de-nitrification. There is also nitrification and aerobic respiration at the top due to availability of oxygen and this phenomenon has been explained in lines Lines 30;  32); 75-78; 246-252 and also confirmed by Heather et al. (2010) and Willian et al. (1986). Nitrification is obligatorily coupled to oxygen consumption and has an effect on the decrease in alkalinity. Such a decrease in alkalinity might cause a decrease in pH (Habboub, 2007). Acidic nitrite formation results in a drop in pH, thus if the buffer capacity of the system is weak, the pH might drop well below 6.7